# Nanomaterials Based Micro/Nanoelectromechanical System (MEMS and NEMS) Devices

**DOI:** 10.3390/mi15020175

**Published:** 2024-01-24

**Authors:** Ziba Torkashvand, Farzaneh Shayeganfar, Ali Ramazani

**Affiliations:** 1Department of Physics and Energy Engineering, Amirkabir University of Technology, Tehran 15875-4413, Iran; zitorkashvand@aut.ac.ir (Z.T.); fshayeganfar@aut.ac.ir (F.S.); 2Department of Mechanical Engineering, Massachusetts Institute of Technology, Cambridge, MA 02139, USA

**Keywords:** micro-electromechanical systems, nanoelectromechanical systems, two-dimensional (2D) materials, sensors, accelerometers, microphones, electronic, optical and mechanical properties

## Abstract

The micro- and nanoelectromechanical system (MEMS and NEMS) devices based on two-dimensional (2D) materials reveal novel functionalities and higher sensitivity compared to their silicon-base counterparts. Unique properties of 2D materials boost the demand for 2D material-based nanoelectromechanical devices and sensing. During the last decades, using suspended 2D membranes integrated with MEMS and NEMS emerged high-performance sensitivities in mass and gas sensors, accelerometers, pressure sensors, and microphones. Actively sensing minute changes in the surrounding environment is provided by means of MEMS/NEMS sensors, such as sensing in passive modes of small changes in momentum, temperature, and strain. In this review, we discuss the materials preparation methods, electronic, optical, and mechanical properties of 2D materials used in NEMS and MEMS devices, fabrication routes besides device operation principles.

## 1. Introduction

The high mechanical flexibility and strain-bearing ability of 2D materials make it easier to tailor their physical characteristics when subjected to external strain, which affects the band structure of crystals, and their electronic properties as a well-established phenomenon [1,2,3]. Suspended 2D nanostructures as a new class of NEMS sensors have attracted great research work due to their unique properties. Thermal isolation besides freedom of motion in suspended nanostructures increase, which enhance mechanical sensing modalities such as high sensitivity and novel functionality in small dimensions. The reason is attributed to the dependence of the sensitivity and performance of NEMS sensors with thickness of the suspended beam or membrane [4]. In recent years, wearable electronics have emerged as a highly prominent research area, with significant dedication directed towards the exploration of high-performance flexible sensors [5]. The creation of the inaugural piezoelectric nanogenerator (PENG) in 2006 [6] followed by the introduction of triboelectric nanogenerators (TENGs) [7]. These innovative devices harness the plentiful and frequently underestimated mechanical energy found in everyday life, encompassing human activities, mechanical operations, water, and wind. Nanoelectromechanical systems have been suggested for extremely sensitive mass detection of neutral species [8]. Substantial advancements have been achieved by employing nanofabricated resonators and carbon nanotubes for mass sensing, as highlighted by the progress in this field [9]. Although graphene is recognized for its exceptional stiffness and strength, the development of graphene-based NEMS devices has some challenges. For instance, the task of applying strains exceeding 1% to graphene while simultaneously measuring alterations in its physical attributes has proven challenging due to the restricted methodologies available for assessing both high strain and other physical properties [10]. In this review, our aim is to provide a comprehensive review of significant advancements in the realm of nanostructure-based nanosensors. While we will touch upon MEMS/NEMS devices, our primary focus will be on electrical, mechanical, and optical sensing, as it holds the potential for facilitating portable assays across a range of point-of-care settings.

## 2. Materials Preparation Methods

In the field of NEMS, researchers pursue the investigation of the potential capability and typical behavior of mechanical devices with nanoscale sizes [11]. They use electrical or optical means to affect the physical motion of specific structures and regulate it. As the device dimensions scale down, the resonant frequency of NEMS increases so the sensitivity and quality factor will improve [12]. Generally, mechanical exfoliation of few-layer flakes from bulk materials is one of the most popular methods in 2D material exploration, where the output is defect-free pure crystalline nanosheets. Although this method facilitates fundamental research on materials properties and the introduction of advanced devices, it is not a reliable process for mass production applications and industrial scales [4]. In the following sections, some preparation methods of materials used in the fabrication of MEMS/NEMS devices are discussed.

### 2.1. Chemical Vapor Deposition (CVD)

For applications that need large area samples such as making multiple arrays, the required materials can be grown using CVD. Synthesis of high-quality layers of 2D materials either directly on the substrate or grown separately to transfer to the substrate for device fabrication is possible in the CVD process. Chemical vapor deposition may be defined as the deposition of a solid on a heated surface from a chemical reaction in the vapor phase [13,14]. In a CVD setup, vaporized chemical materials and inert gas are entered into a quartz tube with controlled temperature and pressure, to form a layer of solid as coating, powders, fibers, and monolithic components. It is possible to grow low-dimensional nanomaterials directly on the device substrate. This process will be followed by device fabrication steps. For example, catalysts such as iron or cobalt nanoparticles, ferritin protein, and thin films of metals can be used to grow nanotubes directly by chemical vapor deposition (CVD). In this method, the nanotube diameter depends on the size of the nanoparticles [11]. The first time synthesized SiNWs (silicon nanowires) were reported by using Vapor-liquid-Solid (VLS) in 1964. In this process, usually, a droplet of metal such as gold (Au), iron (Fe), platinum (Pt), and aluminum (Al) is deposited followed by a subsequent CVD growth of SiNWs [15]. According to Figure 1, First, reactant gases (blue circles) are transported into the reactor (step a). Then, there are two possible routes for the reactant gases: directly diffusing through the boundary layer (step b) and adsorbing onto the substrate (step c); or forming intermediate reactants (green circles) and by-products (red circles) via the gas-phase reaction (step d) and deposited onto the substrate by diffusion (step b) and adsorption (step c). Surface diffusion and heterogeneous reactions (step e) take place on the surface of the substrate before the formation of thin films or coatings. Finally, by-products and unreacted species are desorbed from the surface and forced out of the reactor as exhausts (step f) [16].

### 2.2. Physical Vapor Deposition (PVD)

Physical vapor deposition (PVD) is a widely recognized technology extensively employed for depositing thin films to meet diverse requirements. These demands encompass enhancing tribological behavior, improving optical properties, elevating visual and aesthetic qualities, and various other fields. The technology has already been successfully applied in a broad array of applications, firmly establishing its versatility and utility [17]. PVD stands out as an exceptional vacuum coating method utilized to enhance the resistance against wear and corrosion. This process is particularly crucial for functional purposes, including tools, decorative items, optical enhancements, molds, dies, and blades. These examples merely scratch the surface of the extensive array of applications that have already gained a solid footing in various industries [18,19,20].

### 2.3. Mechanical Exfoliation

Mechanical exfoliation, also known as the “scotch tape method”, or micromechanical cleavage method uses scotch tape as a very simple tool for exfoliation [21]. Using this technique, the isolation of layers can be controlled down to a monolayer, and the flake size depends on how available are the large crystals for exfoliation. This technique can be applied to any bulk-layered crystals whose layers are held together by interlayer weak van der Waals interactions [22,23].

### 2.4. Liquid-Phase Exfoliation (LPE)

Liquid-phase exfoliation (LPE) is a sophisticated technique in materials science that intricately orchestrates the dispersion of layered materials within a solvent milieu. Figure 2 shows the complete process of the LPE [24]. This intricate process unfolds through meticulously mediated exfoliation, wherein the goal is to extract individual layers with a high degree of precision. The exfoliation mechanism is a delicate procedure, relying on the adept overcoming of the formidable van der Waals forces that inherently bind the periodic layers together. This procedure of forces necessitates a nuanced approach to tease apart the layers, transforming the bulk material into finely separated entities, each possessing unique properties and characteristics. Through this methodical manipulation, liquid-phase exfoliation emerges as a powerful tool for tailoring and manipulating the properties of layered materials, unlocking a plethora of possibilities for advanced applications in fields ranging from electronics to nanotechnology [25].

### 2.5. Molecular Beam Epitaxy (MBE)

Molecular Beam Epitaxy (MBE) stands as a cutting-edge methodology for the precise fabrication of thin films within a meticulously controlled high-vacuum environment. In the realm of materials science, this technique is distinguished by its intricacy, involving the deposition of molecular beams laden with essential elements onto a substrate. The deposition unfolds with remarkable precision, employing techniques such as co-deposition and controlled growth through the strategic manipulation of shutters.

In this highly orchestrated process, the molecular beams, akin to artistic strokes on a canvas, are skillfully directed onto the substrate. The result is the creation of epitaxial films, where the deposited material aligns with the crystal lattice of the underlying substrate with an extraordinary degree of conformity. This epitaxial growth is paramount for the production of thin films with superior structural integrity and electronic properties.

The controlled environment of high vacuum in MBE is essential as it eliminates interference from external contaminants, ensuring a pristine and well-defined growth environment. This level of precision and control over the growth conditions makes MBE particularly advantageous for applications in semiconductor device fabrication and emerging technologies, where the structural and electronic characteristics of thin films play a pivotal role in device performance and functionality [26]. During the 1970s and 1980s, there was a significant period of rapid growth and development in the field of MBE. This expansion encompassed various areas such as the growth of II–VI semiconductors [27], III-nitrides [28], the utilization of van der Waals epitaxy for 2D materials [29].

## 3. Characterization of Micro-/Nanostructures

Ideally, scanning electron microscope (SEM) images are of great interest to check the quality and uniformity of Micro-/Nanostructures even up to wafer-scale sizes. Besides, using SEM images, the periodic array of microstructures with different sizes is proved to be the ultimate result of the fabrication process [30]. X-ray diffraction (XRD) is a common tool to determine the quality of crystallinity and crystal properties such as lattice constant and space group. TEM and electron diffraction patterns are ideally used for nanomaterials characterization. Besides, optical spectroscopy together with newly utilized characterization techniques like scanning tunneling microscope (STM) and atomic force microscope (AFM) found their applications [31]. Scanning Probe Microscopy (SPM) represents another third-order nonlinear optical technique that holds the potential to analyze the optical nonlinearity of 2D materials [32,33]. The underlying physical principle of XPS (X-ray photoelectron spectroscopy) and similar photoelectron spectroscopy techniques involves the ejection of electrons from illuminated surfaces (Figure 3). Central to the spectrometer’s operation is the electron energy analyzer, with the electrostatic hemispherical analyzer being the prevalent variant. This analyzer typically comprises two hemispheres positioned concentrically [34,35,36].

## 4. Properties of Micro-/Nanostructures

Micro-/nanostructures exhibit a broad range of applications owing to their fashionable properties. Nevertheless, despite their versatile utility, certain inefficiencies curtail their optimal utilization within specific domains.

### 4.1. Anti-Reflective Properties

One of the best ways to improve the anti-reflective properties of micro-/nanostructures is to make a mixture of microstructures and nanostructures. Repeated reflections and refractions of light are the consequence of a double-scale surface, so the anti-reflectance improves.

### 4.2. Super-Hydrophobic Properties

Hydrophobic materials have attracted great attention in recent years due to their potential application in industrial-level production and basic research [37]. An increase in surface roughness can lead to improvement in the hydrophobicity of low-surface-energy materials. Nanostructures, such as nanoporous, nanowires, and nanopillars, are typical additives to roughen the surfaces of materials to reach super-hydrophobicity [30].

### 4.3. Mechanical Properties

Exploring the mechanical properties of nanostructures are necessary for designing realistic MEMS/NEMS devices [38]. There are various experimental tests such as hardness and elastic modulus tests, fracture toughness tests, scratch resistance tests, and bending tests. Refs. [39,40] and theoretical [41,42] techniques to obtain knowledge of the mechanical properties of nanomaterials. Investigated the electromechanical properties of graphene and its derivatives in the form of graphene oxide and graphene. They reported the highest gauge factor of 300.5 for the hydrogenated graphene oxide and mechanically exfoliated graphene and graphene oxide showed gauge factors equal to 60.3 and 11.2, respectively [43]. The gauge factor values serve as indicators of the potential suitability of these materials for use as force and strain sensors.

## 5. Structural Design

There are various types of designs and architectures for microstructures, depending on the potential applications and material efficiencies. Typically, periodic structures are widely used in many fields as an essential requirement to fulfill the compatibility and uniformity of micro-/nanofabrication [30].

## 6. MEMS/NEMS Fabrication Techniques

### 6.1. Deep Reactive-Ion Etching (DRIE)

Generally, there are two main types of plasma etching processes, including the reactive-ion etching (RIE) process and the deep reactive-ion etching (DRIE) process. In these two processes the plasma is utilized to modify the substrate to make the etching possible; while their working principles are different. The RIE process uses a radio frequency (RF) source to generate the plasma (Figure 4) DRIE is an anisotropic etching process, where a high-concentration plasma ion is utilized for etching, and another plasma gas is used to passivate the surface by a polymeric layer [30].

### 6.2. Nanoimprint Lithography

Nanoimprint lithography (NIL) is a low-cost, high-resolution, and high-throughput technique that was developed in the middle of the 1990s [31,44]. In this method, a predefined stamp which is usually made of a transparent material such as fused silica or polydimethylsiloxane (PDMS) deforms a thermally or UV-cured imprint resist mechanically by pressing. It is crucial that the imprint resist has no deformation or damage; then it can be used either as a direct etching mask or a predefined pattern for the subsequent deposition of metallic materials [45,46].

### 6.3. Nanosphere Lithography

With the nanosphere lithography (NSL) method, patterned masks for uniform nanometric structure etching down to sub-10 nm can be constructed. In this technique, periodic structures of self-assembled closed-pack nanospheres made of specific polymers such as polystyrene or other sphere-shape materials are utilized to prepare a practical nanomesh of substrate materials [47,48]. NSL is a useful method for probing and detecting single molecules in the field of organic electronics and as a means to fabricate highly oriented solar cells [49]. Figure 5, shows the nanosphere lithography step by step.

### 6.4. Electron Beam Lithography (EBL)

Electron beam Lithography (EBL) provides the capability to manipulate nanostructures down to ∼2 nm resolution thanks to a highly focused electron beam exposed on a thin electron-resist layer of polymer. In this process, the solubility of the resist is changed during the exposure time so it will be possible to remove or maintain specific areas of resist by developing, depending on the negative or positive type of etching mask, same as photolithography [50,51].

### 6.5. Photolithography

Photolithography is one of the pioneering techniques for microscale patterning. Originally, it was proposed for mass production in micro-/nanoelectronics and is progressively being used for the large-scale fabrication of optical and mechanical devices. According to Figure 6, a typical photolithography consists of a pattern transfer from a mask during a single exposure step to a photoresistive material. Nevertheless, the use of toxic or dangerous materials together with special ambient conditions restricts the adaption of lithographic fabrication methods and triggers the development of alternative miniaturization technologies and advanced materials [52,53].

## 7. Device Operation Principles and Functionality

Following the successful fabrication of devices, the subsequent essential step involves the comprehensive characterization of these devices. This characterization process entails a thorough examination and analysis of various parameters, functionalities, and performance metrics to gain insights into the behavior and qualities of the fabricated devices.

### 7.1. Optical Detectors

In the typical approach, a high-energy laser (typically at a wavelength of 405 nm) is concentrated on the substance and adjusted according to the operating frequency. Consequently, the cantilever’s temperature is modified, resulting in periodic contraction and expansion of the layer, ultimately causing motion. The movement of the cantilever is observed by monitoring the changes in intensity modulation of the reflected signal through the utilization of a photodiode [54,55]. Koshelev et al. [56] created a nonlinear nanoresonator with a cylindrical shape, reaching a height of 635 nm (Figure 7). The nanoresonator is composed of aluminum gallium arsenide, AlGaAs material, and is positioned on a custom-engineered three-layer substrate consisting of SiO_2_/ITO/SiO_2_. ITO is a glass substrate with indium tin oxide on top making the glass both low resistance and highly transparent. Their research focused on examining individual dielectric nanoresonators that possess a quasi-BIC (Bound-State in the Continuum) resonance at telecommunication wavelengths. Additionally, they demonstrated the remarkable capability of these resonators for second-harmonic generation (SHG) processes.

One of the recently introduced techniques involves utilizing compressive buckling facilitated by a stretched elastomeric substrate to direct the mechanical construction of intricate 3D mesostructures. Some of these structures exhibit designs that recall the achievements seen in larger-scale origami/kirigami structures. These 3D mesostructures are engineered with predefined shapes and can encompass a wide range of sizes, varying by several orders of magnitude in characteristic dimensions. This spans from dimensions in the submicron range for lateral features to thicknesses as small as a few tens of nanometers [57,58].

### 7.2. Electrical Detectors

The performance of devices can also offer crucial insights into the properties of materials. A prime illustration of this is the use of field-effect transistors (FETs), which have emerged as a primary method for assessing the electrical characteristics of 2D materials [33]. To enable rapid and straightforward DNA detection, diverse label-free field-effect transistor (FET) systems have been created. Specifically, thin-film transistors (TFTs), akin to other FETs in their operational principles and structure, offer significant advantages [59,60]. They can be tailored to various insulating substrates (such as glass or flexible materials) and feature a straightforward manufacturing process devoid of additional doping (thus eliminating the need for p-n junctions) [61].

### 7.3. Infrared Detectors

The two key factors for the fabrication of ultrafast high-resolution micro-/nanoelectro-mechanical resonant infrared detectors are the device size scale down, up to the nanometer range and the capability of high infrared absorption in such a comparatively low thickness, remaining the electromechanical achievement untouched. Qian et al. [62] introduced a high-frequency (307 MHz) NEMS resonant IR detector based on graphene/AlN piezoelectric nanoplate. They demonstrated the first-time-exploited improved performance in IR absorption up to 100 times, thermal resistance, and quality factor for reduced volumes. The guaranteed noise equivalent power (NEP) in the order of 1 pW Hz^−1^ is suitable for low-noise multispectral thermal imagers. Although photonic detectors have a high signal-to-noise ratio and fast response time, for canceling thermally generated carriers they need cryogenic cooling. Hui et al. [63] proposed plasmonic piezoelectric NEMS resonant structures for infrared detecting with no need to cool down. These sandwich structures made of aluminum nitride (AlN) piezoelectric nanoplate and platinum (Pt) layers are capable of selectively detecting high-frequency (162 MHz) mechanical vibrations as a step toward the implementation of high-performance and power-efficient infrared/THz imaging systems. Based on Postma prediction [Dynamic range of nanotube- and nanowire-based electromechanical systems], high-aspect-ratio NEMS resonators enter a nonlinear regime as amplitudes of motion x reach a maximum value. Thereafter, they can be accurately described by the Duffing model [64]. Manzeli, et al. [65] investigated the nonlinear dynamic behavior of the NEMS resonators based on suspended monolayer ${\mathrm{MoS}}_{2}$. They showed that the output electrical signal of these ${\mathrm{MoS}}_{2}$ NEMS resonators will enhance due to the presence of piezoresistivity.

### 7.4. Mass Sensors

By using the shift detection in the resonance frequency of the nanotubes, it is possible to sense and evaluate the inertial mass of atoms and nanotubes. In this regard, H. Chiu et al. [12] utilized doubly clamped suspended carbon nanotube nanomechanical resonators for mass sensing in atomic scale applications. This ultrasensitive mass detection would lead to the on-board mass detection, chemical compounds elemental analysis and identification, and materials impurity specification. Figure 8a shows the SEM image of a representative suspended nanotube device with source, drain, and gate electrodes. Figure 8b represents the source-drain conductance versus gate voltage showing Coulomb peaks and single-electron transistor behavior at T ∼ 6 K. Figure 8c shows a schematic diagram of the measurement circuit. The transmission lines with characteristic impedance $Z_{0}$) 50Γ are both terminated at each end with a 50Γ resistor to the ground. The voltages $V_{0},V_{AC},V_{DC}$, and $U_{sd}$ are described in the text. The nanotube is shown in red. Furthermore, Conventional mass spectrometry faces challenges when it comes to measuring the mass of particles in the mega to gigadalton range. Nanomechanical mass spectrometers, as nanomechanical resonators, frequently encounter issues such as significant sample loss, prolonged analysis time, or insufficient resolution. However, Chiu et al. present a system architecture that addresses these challenges. Medina et al. [66] approach involves nebulizing the analytes from a solution, efficiently transferring and focusing them without relying on electromagnetic fields and conducting mass measurements of individual particles using nanomechanical resonator arrays.

### 7.5. NEMS Contact Switches

NEMS switches are of great interest as they have good records of leakage current control and as a consequence lead to lower power consumption and improved ON/OFF ratios, where it is possible to control the current flow between source and drain terminals, using electrical and mechanical tools [65]. The process of design and fabrication of a silicon NEMS resonator is shown in Figure 9. In nanotube-based crossbar memories, two layers of nanotube structures cross by a nanometric gap in which data are WRITTEN by an actuation voltage 2.5 V where the two nanotube ropes make contact. In the case of suspended nanostructure switches, a single nanometric structure is fixed at two ends and suspended over any electrode of a two/three-terminal device. Actuation voltages up to 5 V and sub-3-ns switching times are reported for these structures [67].

### 7.6. MEMS/NEMS Resonant Sensor

Biomarkers have the potential ability the predict, diagnose, and monitor diseases. Typically, miRNA detection techniques are limited due to the difficulty in amplification and the expensive process. Xue et al. [69] proposed an ultrasensitive surface plasmon resonant (SPR) sensor based on antimonene nanomaterials for the efficient label-free detection of cancer-associated microRNA molecules. They particularly selected miRNA-21 and miRNA-155, as two promising candidates for cancer diagnosis (Figure 10). In this study, two prominent factors play a significant role in the high sensitivity of SPR sensors; strong interaction between antimonene and single-stranded DNA and enhanced coupling between the localized-SPR of gold nanorods and propagating-SPR of the gold substrate. Besides, By utilizing pulsed laser illumination to stimulate a colocalized optical mode within the cavity, MacCabe et al. [70] were able to measure internal acoustic modes with sensitivity to single phonons, even at extremely low temperatures in the millikelvin range. This nano-acoustic resonator approach revealed a phonon lifetime of approximately $\tau_{ph,0}∼1.5$ s (characterized by a quality factor Q of approximately $5\times 10^{10}$) and a coherence time of around $\tau_{coh,0}∼130$ microseconds for cavities that are controlled by bandgap.

### 7.7. Triboelectric Nanogenerators

The effect that two different materials possess equal charges but opposite signs after getting in contact is called triboelectrification, which is most significant to a TENG. However, the understanding of the underlying factors responsible for triboelectricity remains unclear. According to classical theories, triboelectricity is attributed to the movement of electrons from one material to another when the two materials come into contact [71]. So far, studies on many applications of TENG have been conducted. There are several different sources of mechanical energy input to supply TENG, such as human activities [72,73,74], natural resources [75], and magnetic/electric-field resources [76] (Figure 11). Furthermore, owing to their exceptional physical and chemical characteristics, two-dimensional (2D) materials like graphene [77,78,79], transition metal dichalcogenides (TMDs) [80], hexagonal boron nitride (h-BN) [81], MXenes [80], and layered double hydroxides (LDHs) have significantly contributed to the progress of TENGs.

### 7.8. Piezoelectric Nanogenerators

Piezoelectricity can be described as the creation of dipole charges due to the application of mechanical stress or conversely. Quartz and Rochelle salts were predominantly employed as the initial materials for these applications [82]. Meng et al. proposed a seamless combination of a thin-film piezoelectric MEMS and an optical metasurface (OMS) based on gap-surface plasmons which resulted in the development of an electrically driven dynamic MEMS-OMS platform. This advanced platform offers meticulous control over both the phase and amplitude modulation of reflected light through fine actuation of the MEMS mirror [83]. Figure 12 shows the 2D wavefront shaping with the MEMS-OMS from Meng et al.’s work [83].

Sample dimensions are the key factors in engineering the device’s capability. As the width and length of the device determine the number of channels and the interaction range, they would have a distinct influence on the NDR performance [84].

### 7.9. Environmental Sensors

Resonant microcantilevers made of silicon (Si) are widely employed as the predominant choice for gas detection applications that demand a Limit of Detection (LOD) at the sub-parts-per-million (ppm) level. Numerous sensors based on resonant cantilevers have been documented and extensively reviewed in the literature. This includes scenarios such as the detection of volatile organic compounds (VOCs) [85,86], humidity sensors [35,87], chemical warfare agents, and explosives [88].

### 7.10. Strain Sensors

Strain engineering emerges as a viable solution to address some limitations of gapless in graphene nanostructures. Despite its remarkable physical properties, graphene encounters a notable drawback for potential use in electronics—its band structure lacks a gap. The ultimate aim of employing this approach is the development of an all-graphene electronic circuit, where the manipulation of strain allows for the control of electron flow through the circuit [89]. Elastic strain engineering (ESE) has arisen as a potent technique for continuously and reversibly modifying physical and chemical characteristics, and it can even introduce novel functionalities into materials with low dimensions. Lately, there have been reports of achieving what’s termed “deep ultra-strength”, surpassing half of a material’s theoretical strength, particularly in certain low-dimensional nanomaterials. This achievement is considered a key enabler for what is known as “deep elastic strain engineering” (DESE) [2,5,90]. Rhenium disulfide (${\mathrm{ReS}}_{2}$) shows an anisotropic piezoresistive effect which can be beneficial for the sensing of multidimensional gestures or strain signals [1]. Thermionic emission and quantum-mechanical tunneling across the barrier result in an impressive giant gauge factor (GF) of approximately 3933 for a strain sensor based on ${\mathrm{SnS}}_{2}$ when exposed to laser illumination [91]. Liu et al. [92] showed that the spatial modulation of the local bandgap in molybdenum disulfide $\left({\mathrm{MoS}}_{2}\right)$ can reach up to 10% through thermomechanical nanoindentation, where can be a good candidate for NEMS strain sensors. Figure 13 shows a successful sample of using ${\mathrm{MoS}}_{2}$ for fabricating an RF NEMS resonator [93].

## 8. Computational Approach

In recent years, higher mechanical sensitivity of MEMS/NEMS sensors can be found with high cost, high power consumption like cantilever-based sensors for selective gas detectors in many different fields: industrial, medical, and environmental sectors. The heated element sensing method and theoretical femtogram resolution mass detection can be used (individual molecules may be measured due to changes in resonant frequency or surface stress of the device), while volatile organic compound sensing as a NEMS application can be achieved with a combination of a functionalization layer [41].

These devices offer increased mobility through a lab-on-chip solution, where a great reduction in energy consumption and the cost is inevitable. Designing these optimal mechanical sensors can be very iterative and expensive. For instance, growing NEMS, etching, and characterizing require advanced facilities and take several months. To achieve a faster time and reduce the number of iterations to optimal design, simulation techniques assist researchers in saving their time in design configuration, testing the feasibility of new sensor design, and reducing materials cost [41]. One of the computational methods is the finite element analysis (FEA) simulations, which model the mechanical properties of MEMS/NEMS sensors and also couples other environmental parameters such as joule heating effects and polarization together [94]. FEA finds approximate solutions of partial differential equations and integral equations. The solution approach in steady-state problems is based on eliminating the differential equation.

In general, FEA by minimizing the energy functional using variational techniques solves for the unknown field quantities. The energy functional includes all the energy associated with the structure. For example, the energy functional of 3-dimensional time-harmonic electromagnetic fields *E* and *H* is written as: [95]
(1)$$\int_{v}\frac{\mu H^{2}}{2}\frac{\epsilon E^{2}}{2}\frac{JE}{2j\omega }dv$$
here, the energy stored in the magnetic and electric fields are represented by the first two terms and the third term indicates the energy supplied or dissipated by the conduction currents.

Another simulation method is finite difference time domain (FDTD) as a full-wave time-stepping procedure, which is based on the Solution of Maxwell’s curl equation: [94]
(2)$$\nabla \times E=-\mu \frac{\partial H}{\partial t}\nabla \times H=\sigma E+\epsilon \frac{\partial E}{\partial t}$$
here *H* and *E* are the vector magnetic and electric fields, and ϵ, μ, and σ are the permittivity, permeability, and conductivity of the medium, respectively.

Another technique is based on the multiresolution Time Domain (MRTD) technique, which implements multiresolution principles and wavelet expansions to discretize Maxwell’s equations. Furthermore, gridding capability in both time and space has been built in MRDT, which can model changes in structure over time and Variable grid capability in position [96].

Also, Warminski [97] analyzed the vibrations of a nonlinear MEMS device that was self and parametrically excited, driven by both external excitations and time-delay inputs. This paper represents the determination of the amplitudes of periodic oscillations analytically by the multiple time scale method (MS) in the second-order perturbation.

In another report, Yi et al. [98] explored the impact of dust on the performance of (MEMS) thermal wind sensors, aiming to assess their practical applications. An equivalent circuit was developed to analyze the temperature gradient influenced by dust accumulation on the sensor’s surface. The proposed model was validated through finite element method (FEM) simulations using COMSOL Multiphysics software [99]. Experimental verification involves the accumulation of dust on the sensor’s surface using two different methods. Furthermore, Muratet et al. [100] introduced a novel methodology for studying the reliability of MEMS (Micro-Electro-Mechanical Systems) based on the concept of ‘virtual prototyping’. The U-shaped micro electrothermal actuator served as a test vehicle for the demonstrated methodology. Using modeling tools such as MatLab [101] and VHDL-AMS, [102] a ‘virtual prototype’ was created. The study applied a best practice Failure Mode and Effect Analysis (FMEA) on the thermal MEMS to investigate and assess failure mechanisms. The reliability characterization methodology predicted the evolution of MEMS behavior based on the number of operation cycles and specific operational conditions.

Zhang et al. [103] developed and produced MEMS thermopile sensors for differential thermal analysis. The differential thermopiles were batch-fabricated with high-density n-type/p-type single-crystal silicon thermocouples using the “microholes interetch and sealing (MIS)” technique. Thanks to the high Seebeck coefficient of single-crystal silicon and the high-density spiral design, the sensors have demonstrated exceptional temperature responsivity of 27.8 mV/°C and power responsivity of 99.5 V/W. This marks a more than four-fold improvement compared to other MEMS thermopile-based differential thermal analysis chips.

## 9. Conclusions and Future Perspectives

We have reviewed several approaches to manufacturing 2D materials-based MEMS/ NEMS devices. MEMS/NEMS applications generally require 2D materials structures with special electronic, optical, and mechanical properties, where these nanostructures are suspended with special growth and transfer processes. Producing single-layer crystalline 2D materials by engineering the substrate and growth process constitutes a major step and complexity in MEMS/NEMS devices. We believe that all potential applications of these devices could become commercially available with a joint effort from academics and industry. The multidomain and interdisciplinary nature of MEMS/NEMS, which includes interacting physical and chemical phenomena, triggers significant game-changing investments by the scientific community and industry.

## Figures and Tables

**Figure 1 micromachines-15-00175-f001:**
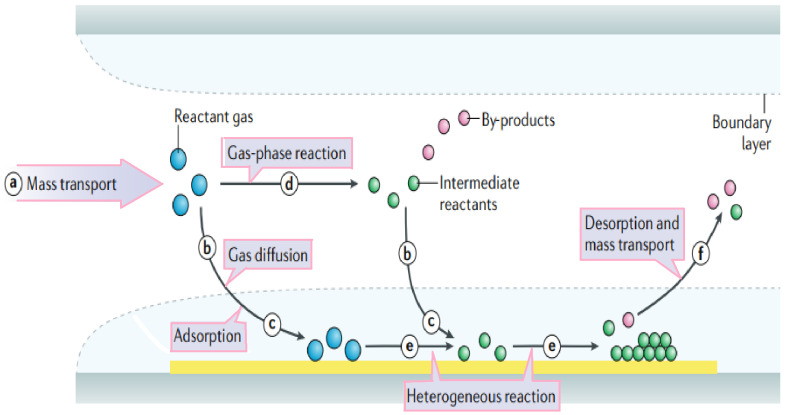
Schematic of general elementary steps of a typical CVD process. Reactant gases (blue circles) enter the reactor (step a). They can either directly diffuse through the boundary layer or adsorb onto the substrate (steps b and c). Alternatively, gas-phase reactions can lead to intermediate reactants (green circles) and by-products (red circles), which are then deposited onto the substrate (steps d, b, and c). Surface diffusion and heterogeneous reactions occur before thin films or coatings form (step e). By-products and unreacted species are desorbed and expelled as exhausts (step f). Adapted from [16].

**Figure 2 micromachines-15-00175-f002:**
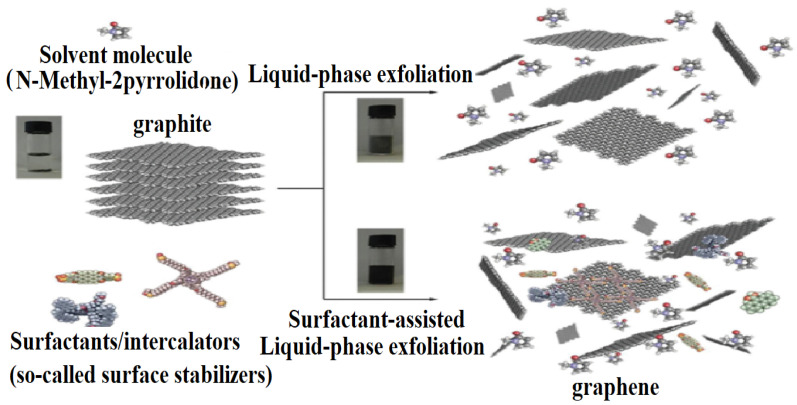
Schematic representation of the liquid-phase exfoliation (LPE) process of graphite in the absence (**top-right**) and presence (**bottom-right**) of surfactant molecules. Reproduced from Ref. [24].

**Figure 3 micromachines-15-00175-f003:**
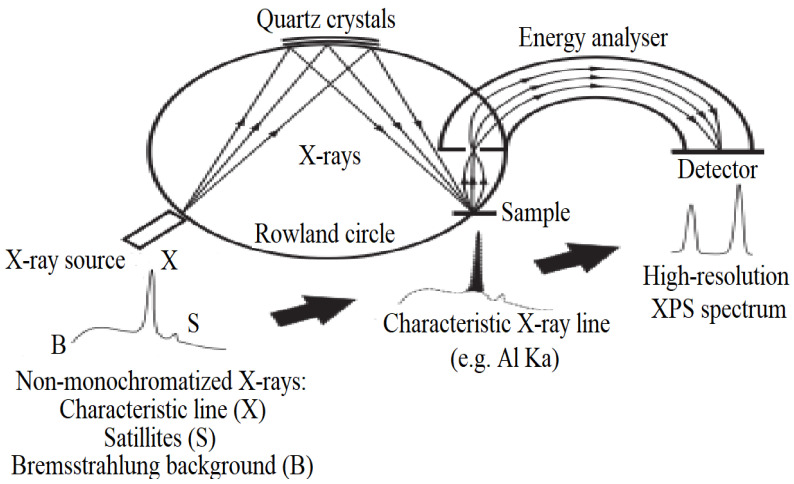
Schematic illustration of the principle behind X-ray monochromatization: X-ray source, a monochromator crystal, and a sample are placed on the circumference of the Rowland circle. Adapted from [36].

**Figure 4 micromachines-15-00175-f004:**
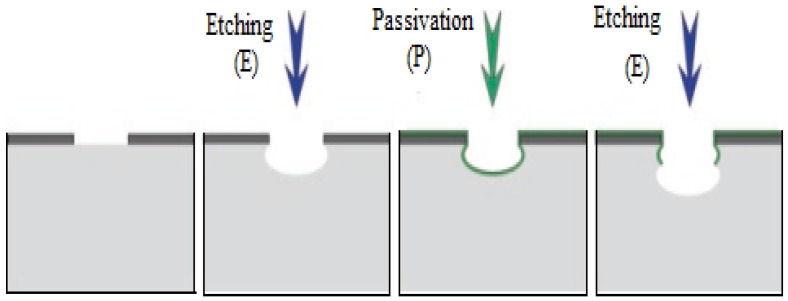
The working principle of standard Bosch DRIE process. In the standard Bosch DRIE process, two types of gases are alternately used in the reaction chamber to realize the etching and passivation steps, respectively. These gases are formed by high-dense plasma ions via glow discharge by RF power. Adapted from [30].

**Figure 5 micromachines-15-00175-f005:**
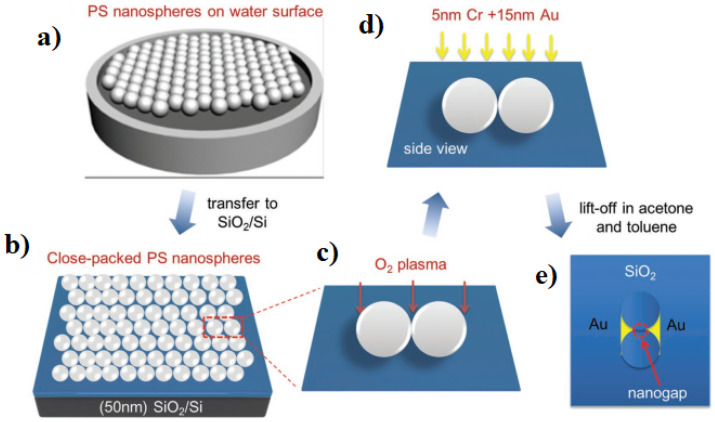
The fabrication of nanogap structures by nanosphere lithography: (**a**) Large-area monolayer of polystyrene nanospheres on the deionized water. (**b**) Monolayer of polystyrene nanospheres transferred onto ${\mathrm{SiO}}_{2}/\mathrm{Si}$. (**c**) $O_{2}$ plasma treatment. (**d**) Metal deposition. (**e**) Lift off the polystyrene nanospheres in the acetone and then toluene to form nanogap. Adapted from [49].

**Figure 6 micromachines-15-00175-f006:**
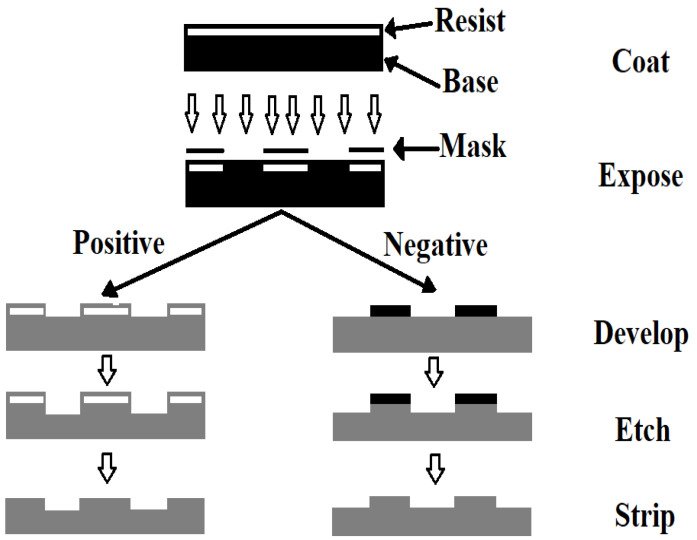
Schematic representation of the photolithographic process sequences, in which images in the mask are transferred to the underlying substrate surface. Adapted from [53].

**Figure 7 micromachines-15-00175-f007:**
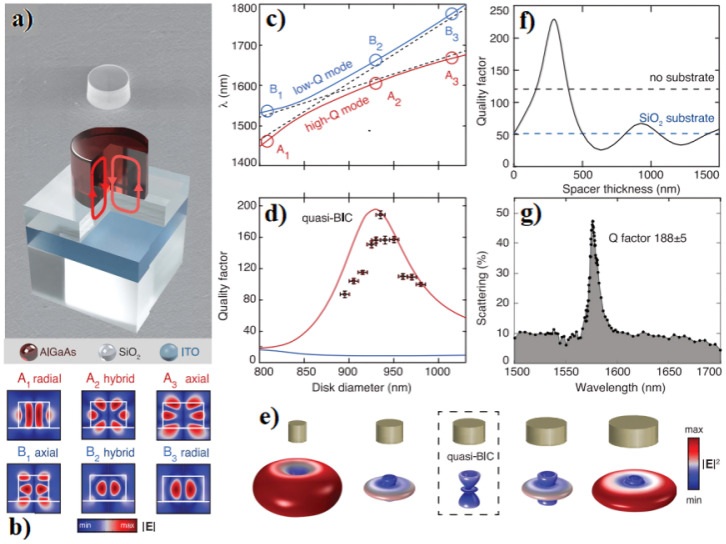
(**a**) Scanning electron micrograph (**top**) and schematic (**bottom**) of an individual dielectric nanoresonator. (**b**) Simulated near-field patterns of the two modes for different diameters. (**c**) Calculated mode wavelengths versus resonator diameter. (**d**) Calculated (lines) and measured (dots) Q factors of modes versus resonator diameter. Calculations in (**c**,**d**) are done for a 350-nm ${\mathrm{SiO}}_{2}$ spacer. (**e**) Simulated far-field patterns of the high-Q mode for disks of different diameters are shown schematically. For calculations, ${\left|E\right|}^{2}$ is normalized to the full mode energy. (**f**) Calculated Q factor of the quasi-BIC versus ${\mathrm{SiO}}_{2}$ spacer thickness compared with the Q factors of a nanoresonator in air and on a bulk ${\mathrm{SiO}}_{2}$ substrate (dashed lines). (**g**) Measured scattering spectrum and retrieved Q factor of the observed resonance for a disk with a diameter of 930 nm. Adapted from [56].

**Figure 8 micromachines-15-00175-f008:**
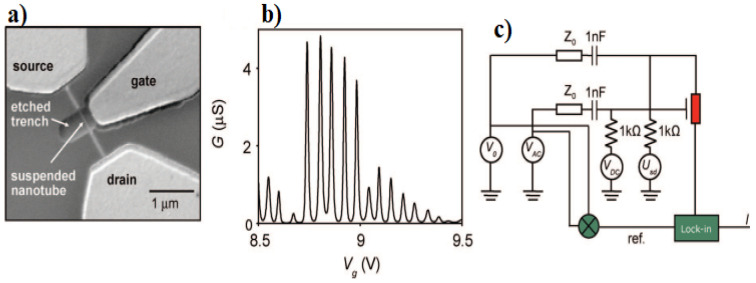
Device geometry of suspended carbon nanotubes, single-electron transistor characteristics, and high-frequency measurement setup. (**a**) Scanning electron microscope image of a representative suspended nanotube device with source, drain, and gate electrodes. (**b**) Source-drain conductance versus gate voltage showing Coulomb peaks and single-electron transistor behavior at T ∼ 6 K. (**c**) Schematic diagram of the measurement circuit. Adapted from [12].

**Figure 9 micromachines-15-00175-f009:**
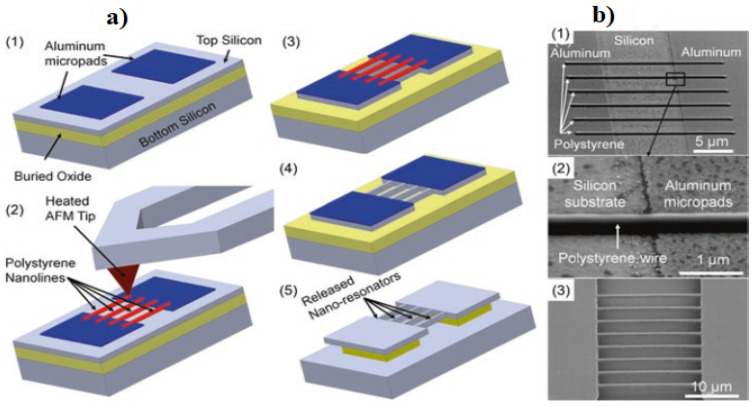
(**a**) The fabrication process of a silicon NEMS resonator using tDPN combined with a regular microfabrication process. (**b1**) SEM images of deposited polystyrene nanolines across aluminum micropads defined by regular microfabrication. (**b2**) Magnified view of polystyrene nanoline across the silicon surface and the aluminum micropad. (**b3**) Suspended silicon nanoresonators produced. Adapted from [68].

**Figure 10 micromachines-15-00175-f010:**
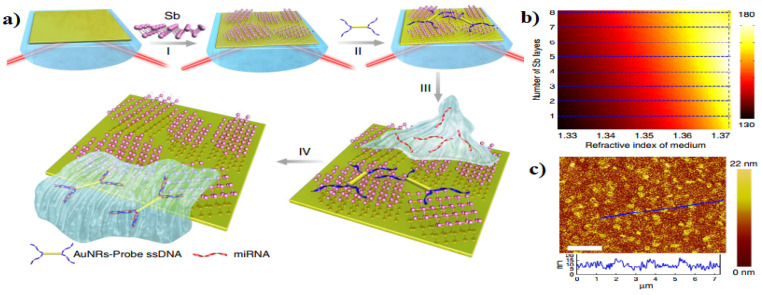
(**a**) Fabrication of miRNA sensor integrated with antimonene nanomaterials. I The antimonene nanosheets were assembled on the surface of Au film. II AuNR-ssDNAs were adsorbed on the antimonene nanosheets. III miRNA solution with different concentrations flowed through the surface of antimonene, and paired up to form a double-strand with complementary AuNR-ssDNA. IV The interaction between miRNA with AuNR-ssDNA results in the release of the AuNR-ssDNA from the antimonene nanosheets. The reduction in the molecular of the AuNR-ssDNA on the SPR surface makes for a significant decrease of the SPR angle. (**b**,**c**) The variation in the sensitivity of the proposed biochemical sensor when the refractive index of the sensing medium is 1.37 + n with respect to the different number of antimonene layers. Adapted from [69].

**Figure 11 micromachines-15-00175-f011:**
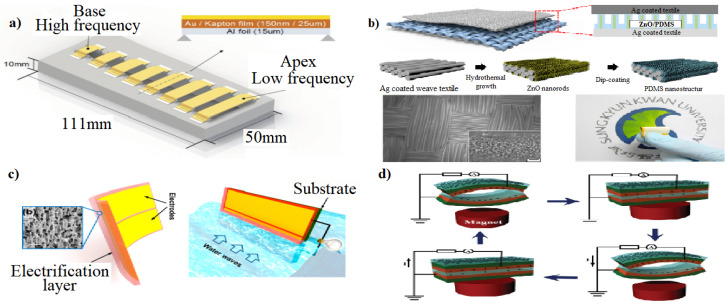
(**a**) Schematic drawing of the TEABM with eight beams. Each beam is a fixed beam with Kapton film and aluminum foil [72]. (**b**) Schematic illustration of the WTNG, Fabrication process of the nanopatterned PDMS structure, FE-SEM images of the bottom textile with nanopatterned PDMS. Inset is a high-resolution image clearly showing the ZnO NR-templated PDMS nanopatterns, a Photographic image of the flexible, foldable WTNG [74]. (**c**) Structural design of the LSEG. Schematic of the bent electrification layer with two electrodes on one side, Schematic of a substrate-supported LSEG positioned in water waves. The up-and-down movement of the surrounding water body induces electricity generated between the two electrodes [75]. (**d**) Original state, Absorption state under the action of a magnet, Release state when removing the magnet, and Absorption state under the action of a magnet. Adapted from [76].

**Figure 12 micromachines-15-00175-f012:**
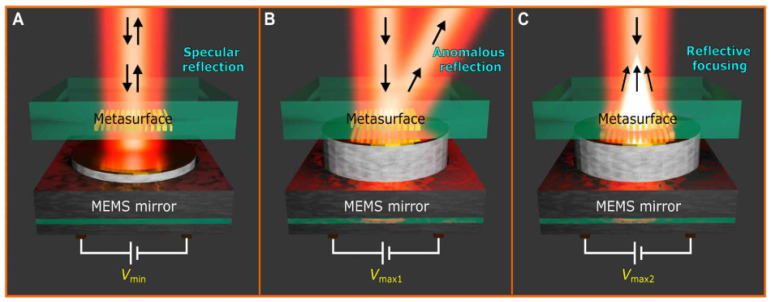
2D wavefront shaping with the MEMS-OMS. (**A**) Schematic of mirror-like light reflection by the MEMS-OMS before the actuation, i.e., with the initial gap of 350 nm between the OMS nanobrick arrays and MEMS mirror. (**B**,**C**) Schematic of demonstrated functionalities, (**B**) anomalous reflection, and (**C**) focusing (depending on the OMS design), activated by bringing the MEMS mirror close to the OMS surface, i.e., by decreasing the air gap to 20 nm. Adapted from [83].

**Figure 13 micromachines-15-00175-f013:**
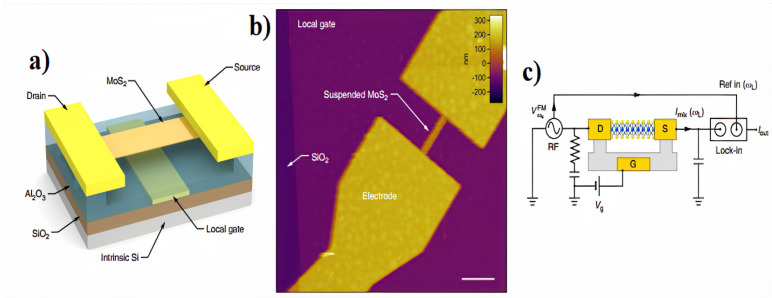
Nanoelectromechanical devices based on monolayer ${\mathrm{MoS}}_{2}$ and the RF electromechanical measurement setup used for characterization of the ${\mathrm{MoS}}_{2}$ NEMS resonators. (**a**) schematic of a monolayer ${\mathrm{MoS}}_{2}$ resonant channel transistor. (**b**) AFM image of a monolayer ${\mathrm{MoS}}_{2}$ ribbon suspended over a local gate electrode and clamped with source/drain electrodes. Scale bar: 1 µm. (**c**) Schematic illustration of the RF electromechanical measurement setup. Adapted from [93].

## Data Availability

The data presented in this study are available on request from the corresponding author.

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
