# Peer review of "Nanomaterials Based Micro/Nanoelectromechanical System (MEMS and NEMS) Devices"

_micromachines, 2024, doi:10.3390/mi15020175_

Round 1
Reviewer 1 Report
Comments and Suggestions for Authors
This review covered the eight materials preparation techniques, the electronic, optical, and mechanical characteristics of the 2D materials used in nine NEMS and MEMS devices, and the fabrication processes. This paper is well-written and I would like to recommend the publication after addressing the following questions:
1)Could the author unify the labeling of subgraphs? a), b), c) were used in Fig.5. A,B,C were used in Fig.7. (a), (b), (c) were used in Fig.8. a,b,c were used in Fig.10.
2) When it comes to the fabrication processes in NEMS and MEMS devices, but there are also some related literatures, which can be appropriately included in the paper. For example, Europhysics Letters, Volume 131, Number 5; Europhysics Letters, Volume 125, Number 2; PNAS. vol. 116, no. 27, 13239–13248.
3) In Section 8, the authors give only the finite element analysis method for electromagnetic fields. Can the authors give more analytical methods for multiphysics fields such as thermal fields, time-varying force fields?
Reviewer 2 Report
Comments and Suggestions for Authors
1. Please, check the abbreviations used in text. Some of them are introduced 2 or 3 times (NEMS, TENG, MEMS, etc.), but others are not introduced at all (SEM, XRD, STM, etc.).
2. Please, take care about the figures. Its sides ratio is deformed in many cases, some of them are larger then the paper size (fig. 1, 11 for example). Also, I find that figure captions are too long. All the explanations that are given as a figure cvaptions might have been said in the entire article text.
3. Check line 46 with "improve12"
4. Line 59. Please, check if the word "quarts" is used correctly there.
5. Lines 93-95. Is the term "dance" used correctly there?
6. Please check line 153. Seems t have several misprints.
7. Line 195. What means (abou=sign) there?
8. Line 223. What is ITO?
9. Line 258. Please, check MHz spelling.
10. Line 355. Please check the formula for Rhenium disulfide.
11. Line 396. What is MRDTD? Is it a misprint in MRTD?
Comments on the Quality of English LanguageDespite some minor misprints English is good. However it seems for me too exquisite and sophisticated for an engineering scientific article. I find it worthful to simplify the language.
Reviewer 3 Report
Comments and Suggestions for Authors
Dear Authors,
in your interesting manuscript, the following points should be added/changed to further improve it:
- It would be less irritating if you used the proper template for the journal you submit to with the required font (and without changing the font for the figure captions) etc.
- According to your abstract, this is a review. So please change "Article" to "Review" above the title. And "e-mail@e-mail.com" is surely not your e-mail address.
- Please insert all figures properly within the normal margins of the pages. Some of them are just slightly too broad, some (e.g. Fig. 11 and Fig. 1) are partly vanishing. Besides, Fig. 11 is stretched, and some other figures (especially, but not only Fig. 3 where the circle is no circle at all) also look stretched. Several figures are not sharp. It is necessary to correctly implement all cited figures and not in such a sloppy way.
- Permissions are only mentioned in a few figure captions. Are all other figures taken from open access publications? If so, it is necessary to add the respective licenses; if not, please add the missing permissions.
- Fig. 1: What does the last (partial) sentence in the caption mean?
- Fig. 4: What is meant with "ething"? Please search for a figure with correct description.
- line 195: "about=sign"?
- Caption of Fig. 5 ff: Please subscript the numbers in SiO2 and O2, ditto throughout the manuscript.
- line 391: What do you mean with "curt" here? Actually it should be "curl".
- Generally, it is necessary to improve the language as well as spelling etc. Regarding references, there are always the spaces before the opening bracket missing, and sometimes the brackets are errouneously set behind a full stop.
Comments on the Quality of English Languagesee above
